# The Association of *TMPRSS6* Gene Polymorphism and Iron Intake with Iron Status among Under-Two-Year-Old Children in Lombok, Indonesia

**DOI:** 10.3390/nu11040878

**Published:** 2019-04-19

**Authors:** Dewi Shinta, Chris Adhiyanto, Min Kyaw Htet, Umi Fahmida

**Affiliations:** 1Southeast Asian Ministers of Education Organization Regional Center for Food and Nutrition (SEAMEO RECFON) Pusat Kajian Gizi Regional Universitas Indonesia, Jakarta 10430, Indonesia; dewishinta.deos@gmail.com (D.S.); kyawhtet@gmail.com (M.K.H.); 2Study Program in Nutrition, Faculty of Medicine, Universitas Indonesia, Jakarta 10430, Indonesia; 3Department of Medical Biology, Faculty of Medicine, Universitas Indonesia, Jakarta 10430, Indonesia; asmarinah.si@gmx.de; 4Faculty of Medicine, Universitas Islam Negeri Syarif Hidayatullah, Jakarta 15412, Indonesia; chrisbiomed@uinjkt.ac.id; 5Sydney School of Public Health, Sydney Medical School, The University of Sydney, Sydney, NSW 2006, Australia

**Keywords:** anemia, iron deficiency, *TMPRSS6*, iron intake, children, Indonesia

## Abstract

Multiple common variants in *transmembrane protease serine 6 (TMPRSS6)* were associated with the plasma iron concentration in genome-wide association studies, but their effect in young children where anemia and iron deficiency (ID) were prevalent has not been reported, particularly taking account of iron intake. This study aims to investigate whether *TMPRSS6* SNPs (rs855791 and rs4820268) and iron intake are associated with a low iron and hemoglobin concentration in under-two-year-old children. The study analyzed the baseline of a randomized trial (NUPICO, ClinicalTrials.gov NCT01504633) in East Lombok, Indonesia. Children aged 6–17 months (*n* = 121) were included in this study. The multiple linear regressions showed that *TMPRSS6* decreased serum ferritin (SF) by 4.50 g/L per copy minor allele (A) of rs855791 (*p* = 0.08) and by 5.00 μg/L per copy minor allele (G) of rs4820268 (*p* = 0.044). There were no associations between rs855791 and rs4820268 with soluble transferrin receptor (sTfR) and hemoglobin (Hb) concentration (rs855791; *p* = 0.38 and *p* = 0.13, rs4820268; *p* = 0.17 and *p* = 0.33). The finding suggests the need for further studies to explore whether the nutrient recommendation for iron should be based on genetic characteristics, particularly for children who have mutation in *TMPRSS6*.

## 1. Introduction

Worldwide, Iron Deficiency Anemia (IDA) affects 24.8% of the total population and the highest prevalence is among under-five-year-old children (47.4%) [1], particularly in their first two years of life. Studies often reported a decline in the nutritional status during the first two years of life and therefore this period is a critical window of opportunity for improving childhood nutrition [2].

Previously, IDA has been known to be associated with dietary and/or other environmental factors such as infection. Recently, several studies have indicated a genetic contribution in the development of ID, i.e., 20–30% of variability in iron concentration is attributable to genetic factors [3]. Genome-wide association (GWA) studies have revealed a number of genetic variants of human and animal genomes which influence iron concentration. Studies in the mutant mouse showed that the *transmembrane protease serrin 6* (*TMPRSS6)* genes caused the loss of the catalytic domain of matriptase-2 that concurrently increased the hepcidin level in the liver. Increased hepcidin level led to a severe microcytic anemia due to low iron absorption in the intestine [4]. Findings of severe IDA in masked mice have led to further investigations on the *TMPRSS6* mutations in humans with iron refractory iron deficiency anemia (IRIDA) [5]. 

Recently, GWA studies identified several single nucleotide polymorphisms (SNPs) of the *TMPRSS6* genes that influence microcytic red blood cell phenotypes. Several SNPs of *TMPRSS6* have been identified to be associated with IDA. Among these SNPs, rs855791 and rs4820268 have the strongest association either on red blood cell indices or iron parameters especially in the Asian population [5]. No previous studies have reported the influence of the *TMPRSS6* gene on iron status indicators in under-two-year-old children. In addition, most studies reporting the influence of genetic factors on the iron status did not take into account iron intake at the same time. Therefore, this study aims to investigate whether minor alleles of the *TMPRSS6* gene at rs855791 and rs4820268 as well as the usual iron intake are associated with low iron and hemoglobin concentrations among under-two-year-old children in East Lombok, where the prevalence of anemia and iron deficiency was reported to be high.

## 2. Material and Method

### 2.1. Study Population

Data for this study were collected at the baseline of a randomized trial of NUPICO (Can Nutrigenetics help explain the Mixed Result on Effect of LCPUFAs and Iron on Child Cognition?) that aims to assess gene-nutrient interaction in explaining effect of LC-PUFAs and iron on cognitive functioning of young children. The NUPICO was registered at ClinicalTrials.gov as NCT01504633. Ethical approval was obtained from the Ethical Commission of Faculty of Medicine, Universitas Indonesia (No. 586/PT.02.FK/ETIK/2011, with addendum No.837/UN2.F1/ETIK/IX/2015). Written informed consent was obtained from the parents or caregivers of the children. The study was conducted in 2012 in the East Lombok district, West Nusa Tenggara province, Indonesia. Breastfed, 12–17 months old children of *Sasak* ethnicity were recruited with the following exclusion criteria: Low birth weight (<2500 g), having congenital disorder, severe anemia (hemoglobin, Hb < 70 g/L) and malaria infection. All subjects with complete data on genotype, iron indicators, and iron intake were included in this analysis (*n* = 121). This sample size met the minimum sample (*n* = 94) which was calculated based on the coefficient correlation of iron intake and Hb (r = 0.233, *p* < 0.05) [6] with 0.05 significance level and 80% power. 

### 2.2. Dietary Iron Intake Assessment

The usual iron intake of the children was assessed using a validated semi-quantitative food frequency questionnaire (SQ-FFQ) of the previous month intake. The SQ-FFQ represented 90% of iron intake of the study population and was validated using two-day non-consecutive 24 h dietary recalls and showed good correlation (*p* < 0.05) and no significant mean difference in the estimated usual iron intake. The proportion at risk of inadequate iron intake was estimated from two-day non-consecutive 24 h dietary recalls using the IMAPP Version 1.0. 

### 2.3. Iron and Hemoglobin Measurements

The concentration of Iron was assessed by a soluble serum transferrin receptor (sTfR) and serum ferritin (SF), which were analyzed using the Immunosorbent assays (ELISA) method. To correct the influence of subclinical inflammation which was defined by the C-reactive protein (CRP) (>5 mg/dL) and α-1 Acid Glycoprotein (AGP) (>1 g/dL), correction factors were used to adjust the SF concentration following the Thurnham model, i.e., 1.0 (healthy stage), 0.77 (incubation stage, raised CRP only), 0.53 (acute inflammation, raised CRP and AGP), 0.75 (chronic inflammation, raised AGP only) [7]. The concentration of Hb was analyzed using the cyanmethemoglobin method. The body iron store was calculated using SF and sTfR following the method by Cook et al. (body iron mg/kg = −[log(R/F ratio)−2.8229]/0 [8]. Anemia was defined by Hb < 11 g/dL, ID by sTfR concentration > 8.5 mg/L, and/or SF < 12 μg/mL, whereas IDA was defined by a combination of both anemia and ID.

### 2.4. Genotyping Procedure

DNA extraction was done using the Gene Aid Kit (#GB300, Taiwan, Republic of China). The SNP genotyping analysis was done to identify the Allele type of rs855791 and rs4820268 as the *TMPRSS6* gene variants. The method used Taqman-assay with two allele-specific probes and a pair of primers to specifically detect the type of genotype/allele, i.e., the Applied Biosystem TaqMan SNP Genotyping Assay ID C_11885329_10 for rs855791, ID C_32899902_10 for rs4820268 and GTX press Master mix kits (Cat. 94404, New York, NY, USA) kits. The Step One^TM^ Real Time PCR-system with a 48-well thermo bloc instrument and the stepOneV2.3 software (AB) was used for the analysis of SNPs genotyping assay. A representative sample (twelve replicates) was repeated to confirm the genotype and the success rate was 100%.

### 2.5. Statistical Analysis

Statistical analyses were performed using the SPSS statistical software for Windows (version 20.0; SPSS, Inc., Chicago, IL, USA). Continuous variables (iron intake, sTfR, SF and Hb) were tested for its normal distribution using the one-sample Kolmogorov-Smirnov test. Distributions of continuous variables were presented as mean ± SD (for normally distributed data) and median (25th to 75th) (for non-normally distributed data). Polymorphism of the *TMPRSS6* gene, SNP rs855791 and rs4820268 were presented as a percentage. Multiple linear regression analyses were conducted separately between SNP rs855791 and rs4820268 because of existing multicollinearity between the two SNPs, as indicated by the value of variants inflation factor (VIF) > 1/(1-R^2^) and tolerance (TOL) < (1-R^2^). 

## 3. Results

Table 1 showed that the homozygote minor allele frequency (MAF) genotype (AA) of rs855791 and (GG) of rs4820268 were the biggest proportion of their genotype profile. The Hardy-Weinberg Equilibrium (HWE) test showed that the (q) variant allele frequency of rs855791 (A) and rs4820268 (G) were 74% and 71%, respectively. Whereas the (p) wild-type allele frequency of rs855791 (G) and rs4820268 (A) were 26% and 29%. Two-thirds (67.7%) of the children were also found to be at risk of having inadequate iron intake. 

Based on data analysis from a total of 121 children, 69.4% suffered from iron deficiency, 92.6% had anemia and 63.6% had iron deficiency anemia. We found an increasing trend of iron deficiency, anemia and iron deficiency anemia in children who had minor allele (A) as iron lowering allele (ILA) in rs855791 and minor allele (G) in rs4820268 (Figure 1 and Figure 2).

There were comparable characteristics in terms of sex, age, and the usual iron intake from complementary feeding between genotypes of rs855791 and rs4820268, with the exception of AGP by genotype of rs4820268 (*p* < 0.05), in which AGP is lower with more (G) allele in this SNP (Table 2).

The multiple linear regressions showed that after controlling for the usual iron intake, sex and age, per copy minor allele (G) was significantly associated with lower SF by 5.00 μg/L of rs4820268 (*p* = 0.044) and borderline significantly associated with lower SF by 4.50 μg/L of rs855791 (*p* = 0.080) (Table 3).

## 4. Discussion

Our study reported the association of the *TMPRSS6* genes to the hemoglobin and iron status indicators in young children in the area where IDA is prevalent. To our knowledge, this is the first study that reported such an association in young children (<24 months. We also included simultaneously both genes and iron intake in our analysis, which had not been reported in previous studies. The important finding of this study was that polymorphism of the *TMPRSS6* gene SNP rs855791 and rs4820268 and iron intake both had an association with the iron status, particularly in the SF concentration.

The study population came from a specific tribe in Lombok Island, Indonesia (*Sasaknese)*, which is categorized as a close population. We found the HWE calculation, p + q = 1 which means that both allele (A) of rs85571 and (G) of rs4820268 fulfill the equilibrium condition [9,10].

Our study showed that the minor allele of TMPRSS6 rs4820268 (G) and rs8559791 (A) were associated with a lower SF but not with sTfR, Hb and body iron store. Serum ferritin reflects the iron status indicator, which is sensitive to the early stage of iron deficiency, i.e., iron depletion stage, whereas sTfR reflects a later stage of iron deficiency [11,12]. In the development of ID, sTfR may still be in the normal concentration during the iron depletion stage (during which SF already decreased) and iron-deficient erythropoiesis (during which serum iron and transferrin saturation start to decrease) [13].

This study was done within a specific ethnicity in Indonesia (*Sasaknese*) and did not represent the multi-ethnicity population of Indonesia. We consider *Sasaknese* as an example of a highly prevalent IDA population; and while this is a close population, the genotype data is typical of Asians. The Chinese Han population study by Gan et al. and another study by Chamber et al., *l* which include SNP rs855791 and rs4820268 of the *TMPRSS6* genes showed that minor allele frequency (MAF) of both SNPs in Asia was comparable with the finding in our study population (>55%) [5,14]. There is a considerable discrepancy in the minor allele frequency distribution of both SNPs among the Asian, African and European population [5,14,15]. The minor allele frequency of SNP rs855791 and rs4820268 tend to be lower in African than European and Asian populations. This difference may be caused by certain environmental conditions acting through selective pressure to alter the frequency of the genetic variants among population [16]. Our finding on the variant allele frequency showed that the frequency of variant allele in both SNPs among *Sasaknese* was higher than the study in the Chinese Han and Indian Asian population in which variant allele “A” of rs855791 and variant allele “G” of rs4820268 was within range 50–56% [5,14]. This condition could have occurred because *Sasaknese* had a tradition to maintain their lineage through marriage with the same *Sasaknese.* Thus, the probability of variant allele to appear in the next generation was higher than random mating.

Our finding showed that iron intake had a more significant association as compared to polymorphism of rs855791 and rs4820268 with SF concentration. These results showed that while both gene (nature) and intake (nurture) had an association with the iron status, nurture had a more significant role to the iron status in human. Tanaka et al. found that 20–30% of the variability in iron concentration is attributable to reduced activity of the *Matriptase-2* protein; the latter is determined by the environment including the dietary iron intake [3]. This finding is similar with the study in Tanzanian children, which found that the daily iron intake had a positive correlation to the SF level (r = 0.233, *p* < 0.05) [6]. A similar finding was found on a study among Indian children 12–23 months that according to the multiple linear regression analysis, ferritin was positively associated with iron intake (crudeβ = 0.25 [0.02–0.12]; *p* < 0.01) [17]. Nonetheless, in our study, the variant allele of rs855791 (A) and rs4820268 (G) were associated with a lower SF level even after controlling for iron intake. 

Previous mutation on *TMPRSS6* has been shown to result in the *Matriptase-2* protein lacking activity, which is essential for adequate iron uptake to prevent iron deficiency and to suppress hepcidin expression. Hepcidin is a key iron regulator, which governs systemic iron homeostasis by binding it to ferroportin on the surface of macrophages, enterocyte, and hepatocyte inducing the degradation of ferroportin thereby preventing the efflux of iron in the blood [3,14,15]. In the present study, *TMPRSS6* showed that per copy of the minor allele (G) of rs4820268, the SF concentration reduced by 5.00 μg/L (*p* = 0.044). Meanwhile per copy of the minor allele (A) of rs85579, the SF concentration reduced by 4.50 μg/L (*p* = 0.08). A reduction of 1 μg/L in SF is considered significant as it corresponds to 8–10 mg body iron, while the total body iron store in humans is around 1000–1200 mg [18]. This means that those who have a minor homozygote allele genotype (AA) of rs855791 and (GG) of rs4820268 are naturally depleted by approximately 10% of the total body iron store as compared to the wild-type homozygote. This finding may suggest for the sub-population specific strategies to address iron deficiency, i.e., whether populations with a high prevalence of risk alleles may require higher nutrient recommendations for iron to maintain normal erythropoiesis. This is especially prudent where the high prevalence of risk alleles is combined with the high inflammation burden.

The limitation of our study was that we did not assess hepcidin. We hypothesized that hepcidin is lower in homozygote wild-type as found in Chinese women study where homozygote wild-type (GG) of rs855791 had a lower hepcidin concentration than homozygote minor allele genotype (AA) in the general population and therefore the reduction of the SF concentration by minor allele of rs855791 and rs4820268 was due to over expression of hepcidin [14]. On the other hand, another study by Guo et al. examined the correlation between the hepcidin concentration and iron status indicators and reported that there was a significant positive correlation between serum hepcidin and ferritin concentration among the Chinese Han Population [19]. Another limitation of our study is a small sample size in terms of the interaction between gene and iron biomarkers (statistical power ~70% based on correlation coefficient of 0.38 in SF with significance level of 0.05), therefore we acknowledge the need for a study with a bigger sample size to determine the exact nature of the interaction. Nevertheless, we managed to analyze simultaneously the influence of both gene and intake, which has not been reported before.

## 5. Conclusions

In summary, our study showed that amongst these under-two-year-old children, where anemia and iron deficiency are prevalent, variant allele (A) and (G) at rs855791 and rs4820268 of the *TMPRSS6* gene respectively were associated with a lower SF concentration; but the association was still weaker than that of iron intake. A further study is needed to investigate whether a higher nutrient recommendation for iron based on genetic characteristics, particularly for children who have mutation in *TMPRSS6* will benefit their iron status. 

## Figures and Tables

**Figure 1 nutrients-11-00878-f001:**
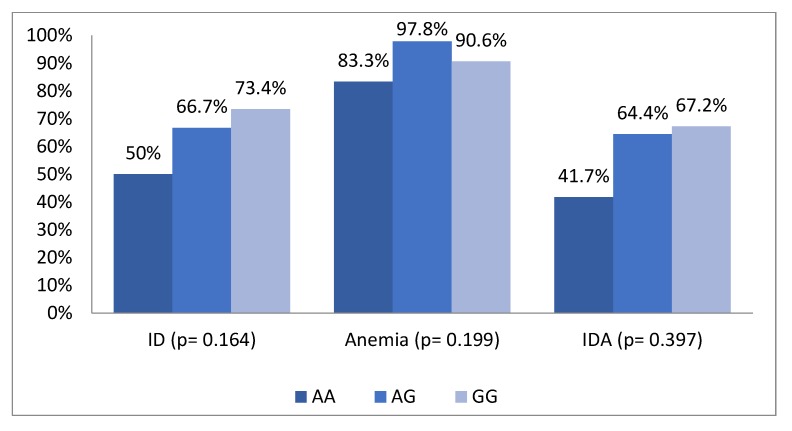
Distribution of Iron deficiency, anemia and iron deficiency anemia status across genotypes of SNPs rs4820268.

**Figure 2 nutrients-11-00878-f002:**
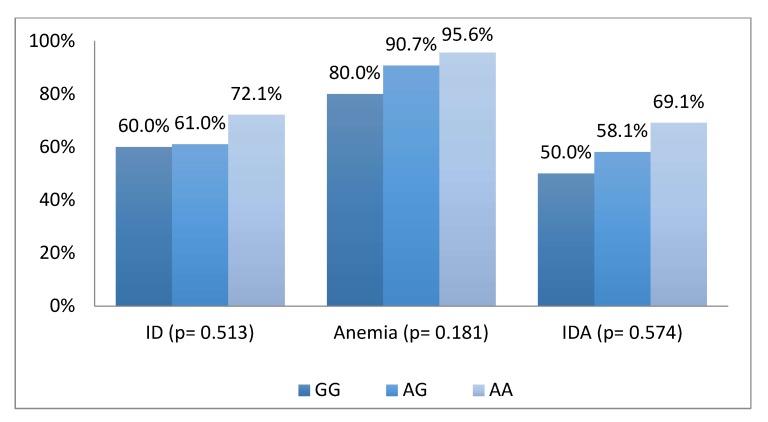
Distribution of Iron deficiency, anemia and iron deficiency anemia status across genotypes of SNPs rs855791.

**Table 1 nutrients-11-00878-t001:** Characteristics of the study population (*N* = 121).

Factors	*n* (%)
**Demographic**	
Age of children * [month]	14.12 ± 1.39
**Sex of children**	
Boy	58 (47.9)
Girl	63 (52.1)
**Maternal Education**	
Primary school	81 (67.0)
Secondary school	40 (37.0)
**Maternal Occupation**	
Not working	88 (72.7)
Working	33 (27.3)
**Genotype Distribution**	
Polymorphism of rs855791	
GG homozygote	10 (8.30)
GA heterozygote	43 (35.50)
AA homozygote	68 (56.20)
Polymorphism of rs4820268	
AA homozygote	12 (9.90)
AG heterozygote	45 (37.20)
GG heterozygote	64 (52.90)
**Minor Allele Frequency (MAF)**	
SNP rs855791 [A/G]	179 (73.90)
SNP rs4820268 [G/A]	173 (71.40)
**Intake**	
Iron [mg/day] *	4.5 ± 3.01
**Iron indicator**	
Serum ferritin [μg/L] *^,†^	14.7 ± 18.00
Serum transferrin receptor [mg/L] ^*^	8.7 ± 3.03
Hemoglobin [g/dL] ^*^	9.51 ± 1.03
Body iron store [mg/kg] ^*^	−0.9 ± 3.8
Anemia	112 (92.6)
Iron deficiency	84 (69.4)
Iron Deficiency Anemia	77 (63.6)

* The data was presented as mean ± SD. ^†^ Serum ferritin was adjusted by subclinical inflammation using Thurnham method as specified in the method section.

**Table 2 nutrients-11-00878-t002:** Characteristics and iron status of the subjects, stratified by genotypes of SNPs rs855791 and rs420268.

	SNP rs855791				SNP rs420268			
	GG (*n* = 10)	GA (*n* = 43)	AA (*n* = 68)	*p*	AA (*n* = 12)	AG (*n* = 45)	GG (*n* = 64)	*p*
Age [mo]	14.08 ± 1.4	14.19 ± 1.3	14.1 ± 1.5	0.945	14.3 ±1.3	14.1 ± 1.2	14.1 ± 1.5	0.875
Sex [boy] *	6 (5)	18 (14.9)	34 (28.1)	0.823	5 (4.1)	21 (17.4)	32 (26.4)	0.812
Iron intake [mg]	4.26 ± 3.17	4.7 ±3.62	4.49 ± 2.57	0.877	3.8 ± 2.9	4.7 ± 3.5	4.5 ± 2.5	0.638
CRP [mg/L]	5.7 ± 7.3	4.4 ± 8.4	3.4 ± 5.8	0.568	6.7 ± 10.1	3.9 ± 7.2	3.5 ± 5.9	0.351
AGP [mg/L]	1 ± 0.3	0.9 ± 0.3	0.8 ± 0.2	0.072	0.9 ± 0.1	0.8 ± 0.3	0.7 ± 0.2	0.049
Serum ferritin [μg/L] ^†^	11.4 ± 7.7	13.9 ± 14.9	10.5 ± 9.7	0.314	12.1 ± 8	14.1 ± 6	10 ± 7.7	0.197
Serum transferrin receptor [mg/L]	7.4 ± 2.5	8.8 ± 2.9	8.8 ± 3.1	0.839	7.4 ± 2.3	8.7 ± 2.8	8.9 ± 3.2	0.290
Body iron store [mg/Kg]	0.5 ± 3.5	−0.4 ± 4.0	−1.4 ± 3.8	0.180	−1.5 ± 3.6	−0.4 ± 4.1	−0.7 ± 3.5	0.123
Hemoglobin [g/dL]	10.04 ± 1.07	9.53 ± 1.08	9.41 ± 0.97	0.195	10.08 ± 1.05	9.37 ± 0.90	9.50 ± 1.08	0.103

* Sex (boy) was presented in *n* (%), ^†^ Serum ferritin (SF) was adjusted by. subclinical inflammation using Thurnham method as specified in the method section.

**Table 3 nutrients-11-00878-t003:** Association between selected SNPs of *TMPRSS6* with Serum Ferritin (SF), Soluble Transferrin Receptor (sTfR), Hemoglobin (Hb) and body iron store.

Factors	Serum Ferritin (μg/L) *			Serum Transferrin Receptor (mg/L)			Hemoglobin (g/dL)			Body Iron Store (mg/Kg Body Weight)		
	β	SE	*p*	β	SE	*p*	β	SE	*p*	β	SE	*p*
**Model I**												
rs855791 (G/A) **	−4.495	2.548	0.080	0.378	0.428	0.379	−0.232	0.146	0.133	−0.213	0.543	0.064
Iron Intake (mg)	1.476	0.551	0.009	−0.109	0.093	0.241	−0.018	0.031	0.565	0.059	0.104	0.518
**Model II**												
rs4820268 (A/G) ^†^	−5.003	2.453	0.044	0.567	0.412	0.171	−0.137	0.142	0.335	−0.067	0.104	0.458
Iron Intake (mg)	1.518	0.549	0.007	−0.114	0.092	0.221	−0.017	0.032	0.598	−0.192	−1.111	0.745

Multiple Linear Regression method = enter (*N* = 121), All factors were adjusted by age and sex of children, Model 1: SF; R^2^ = 0.064, sTfR; R^2^ = 0.002, Hb; R^2^ = 0.009, Model II: SF; R^2^ = 0.072, sTfR; R^2^ = 0.011, Hb; R^2^ = −0.023, * Serum ferritin (SF) was adjusted by CRP >5 mg/L and AGP >1 mg/L, ** rs855791 GG = 0/AG = 1/AA = 2, ^†^ rs4820268 AA = 0/AG = 1/GG = 2.

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
