# Peer review of "The Association of TMPRSS6 Gene Polymorphism and Iron Intake with Iron Status among Under-Two-Year-Old Children in Lombok, Indonesia"

_nutrients, 2019, doi:10.3390/nu11040878_

Round 1
Reviewer 1 Report
This study investigates the association of two SNPs of the TMPRSS6 gene and iron intake with iron deficiency and anemia in a cohort of under two Sasaknese children.
The manuscript needs to be improved in Introduction, Material and Method and Results sections, whereas discussion is well conducted.
Major points:
If rs855791-A allele is the minor allele, why rs855791 AA genotype is the most represented genotype? The same for rs4820268-G allele and rs4820268 GG genotype. Although authors well discuss these results, considering the ethnicity and traditions of the studied population, genotyping mistakes cannot be excluded by repeating TaqMan assay for a representative sample. Random samples must be sequenced to verify. Moreover, authors do not provide any direct info regarding HWE. This is of outmost importance. Is the cohort in Hardy Weinberg Equilibrium? Please add to results and discussion.
Figure 1
Since Table 1 shows number of patients with Anemia or ID or both anemia and ID, why figure 1 shows only stratification of genotypes for Anemia and for ID? Please, add genotype stratification for IDA.
Table 3
Table 3 is not adequately described. The effect of the iron intake seems to be prevalent with respect that of the studied TMPRSS6 SNPs, but is there the possibility to stratify children in two (or more) subgroups with respect to a cut-off (or ranges) for iron intake to get information about the response of the different genotypes to similar iron intake?
If hepcidin has not been evaluated, conclusions regarding the association with TMPRSS6 SNPs must be toned down (p-values do not support a significant association, although the speculation of authors can be agreed).
Minor points:
Table 1:
- n(%) cannot be generalized for all data
- What does it mean Energy? Please specify in Material and method section
- There are mistakes in numbers (i.e. protein, serum transferrin receptor)
Please revise Table 1.
Author Response
Reviewer 1
1. If the rs855791-A allele is the minor allele, why r5855791 AA genotype is the most represented genotype? The same for rs4820268-G Allele and rs4820268 genotype. Although authors well discuss these results, considering the etnicity and traditions of the studied population, genotyping misatkes cannot be excluded by repeating TaqMan assay for a representative sample. Random samples must be sequenced to verify. Moreover, authors do not provide any direct info regarding HWE. This is of outmost importance. Is the cohort in Hardy Weinberg Equilibrium? Please add to results and the discussion.
Re: Based on the following literatures, A is a minor allele of SNP rs855791 (https://www.ebi.ac.uk/gwas/variants/rs855791) (https://www.ncbi.nlm.nih.gov/pubmed/23222517?dopt=Abstract) (https://www.ncbi.nlm.nih.gov/pubmed/20139978?dopt=Abstract) (https://www.ncbi.nlm.nih.gov/pubmed/19862010?dopt=Abstract) (https://www.snpedia.com/index.php/Rs855791) (http://www.bloodjournal.org/content/118/16/4459?sso-checked=true) .
However, minor allele frequencies (MAF) in a country or ethnicity can be a major allele in other ethnicities. We used TaqMan assay which has been designed specifically to detect the targeted allele.
Our calculation based on Rodriquez et al (2009) using the Web program ( http://www.oege.org/software/hwe-mr-calc.shtml) gave the following results:
Result (Rs855791)
X2 = 0.72
(121 samples counted)
for likelihoods of calculated X2 value see below.
Genotype | Expected | Observed |
Common homozygotes | 8.2 | 10 |
Heterozygotes | 46.6 | 43 |
Rare homozygotes | 66.2 | 68 |
p allele freq = 0.26; q allele freq = 0.74
Solutions for perfect HWE, under a model of ascertainment (+/-) of one group
Group affected | Common Hz | Heterozygotes | Rare Hz | p allele freq | q allele freq |
Common Hz | 6.8 | 43 | 68 | 0.24 | 0.76 |
Heterozygotes | 10 | 52.15 | 68 | 0.28 | 0.72 |
Rare Hz | 10 | 43 | 46.23 | 0.32 | 0.68 |
Result (Rs420268)
X2 = 0.93
(121 samples counted)
for likelihoods of calculated X2 value see below.
Genotype | Expected | Observed |
Common homozygotes | 9.84 | 12 |
Heterozygotes | 49.33 | 45 |
Rare homozygotes | 61.84 | 64 |
p allele freq = 0.29; q allele freq = 0.71
Solutions for perfect HWE, under a model of ascertainment (+/-) of one group
Group affected | Common Hz | Heterozygotes | Rare Hz | p allele freq | q allele freq |
Common Hz | 7.91 | 45 | 64 | 0.26 | 0.74 |
Heterozygotes | 12 | 55.43 | 64 | 0.3 | 0.7 |
Rare Hz | 12 | 45 | 42.19 | 0.35 | 0.65 |
Ref: Santiago Rodriguez, Tom R. Gaunt and Ian N. M. Day. Hardy-Weinberg Equilibrium Testing of Biological Ascertainment for Mendelian Randomization Studies. American Journal of Epidemiology Advance Access published on January 6, 2009, DOI 10.1093/aje/kwn359
2. Figure 1, since Table 1 shows number of patients with anemia or ID or both anemia and ID, why figure 1 shows only stratification of genotypes for anemia and for ID? Please, add genotype stratification for IDA.
Re: Thank you for the suggestion; we already add genotype stratification for IDA in figure 1&2
3. Table 1, n(%) cannot be generalized for all data
Re: We have now put either n(%), mean ± SD or Median (25th, 75th) in table heading
4. What does it mean energy? Please specify in material and method section
Re: Energy and protein intakes were estimated from 2-day non-consecutive 24-hour dietary recalls which were used to validate the semi-quantitative food frequency questionnaire (SQ-FFQ). However we consider these to be less relevant to be presented and therefore have deleted from table 1 the energy and protein intakes and only keep the usual iron intake.
5. There are mistakes in numbers (i.e protein, serum transferrin receptor) in Table 1
Re: This has been revised.
Reviewer 2 Report
Comments:
The authors studied the influence of two TMPRSS6 SNPs and iron intake on iron status of under-two-year-old children in Indonesia, providing helpful information on the iron status and iron supplementation in a population with these two TMPRSS6 SNPs.
Issues:
1. Soluble TfR is an indicator of erythropoiesis but not a good indicator of systemic iron status since soluble TfR comes from erythroid cells. In the clinic, Transferrin saturation (serum iron) combining with serum ferritin (iron storage) is the indicator of systemic iron status, should have been used in the study.
2. There are a few mistakes in the paper. Line 23, the unit of serum ferritin should be microgram/L. Line 110, the percentage of children with iron deficiency is 69.4% according to the table, not 68.6%, please verify the data. Page 4, table 1, please verify the number of "serum transferrin receptor".
3. Line 159-161, the explanation about the difference of those two TMPRSS6 SNPs between Chinese and Indian Asian populations is not necessary.
4. Line 40-42, 175-177, these two sentences are not correct in grammar and should be revised.
Author Response
Reviewer 2
1. Soluble TfR is an indicator of erythropoiesis but not a good indicator of systemic iron status since soluble TfR comes from erythroid cells. In the clinic, Transferrin saturation (serum iron) combining with serum ferritin (iron storage) is the indicator of systemic iron status, should have been used in the study.
Re: In this study we assessed serum ferritin and soluble TfR as indicator of iron status rather than Transferin saturation (serum iron). So we generate iron deficiency (ID) status using these two indicators (SF and sTfR). Using the formula recommended by Cook, we have calculated body iron store which can better represent the body iron status. Serum ferritin represents tissue iron status where transferrin receptor represents cellular level iron deficiency and by combining these two indicators, body iron status can better represent iron status of the body.
· CookJD, SkikneB and BaynesR (1996). The use of transferrin receptor for the assessment of iron status. In: Hallberg L. A., Asp N-G, (eds). Iron nutrition in health and disease. London: John Libbey& Co. 91-99.
· Okafor, Mifeyinwa, Antai, Batim, Dokpokan, AEUsanga. Soluble transferrin receptor as a marker in the diagnosis of iron deficiency anemia in pregnancy : A study in Calabar, Caalabar, Nigeria. Int J of Bio Lab Sci (IJBLS). 2014. (3): 2. 41-47 p.
2. There are a few mistakes in the paper. Line 23, the unit of serum ferritin should be microgram/L. Line 110, the percentage of children with iron deficiency is 69.4% according to the table, not 68.6%, please verify the data. Page 4, table 1, please verify the number of
"serum transferrin receptor".
Re: This has been revised.
3. Line 159-161, the explanation about the difference of those two TMPRSS6 SNPs between Chinese and Indian Asian populations is not necessary.
Re: We put this discussion to show how distribution of these SNPs in our study population compared with the Chinese and Indian studies which are from Asian population.
4. Line 40-42, 175-177, these two sentences are not correct in grammar and should be revised.
Re: This has been revised.
Reviewer 3 Report
This short manuscript addresses an issue about transmembrane protease serine 6 (TMPRSS6) in young children particularly where anemia and iron deficiency (ID) were prevalent has not been reported particularly together with iron intake. This study aims to investigate whether TMPRSS6 SNPs 18 (rs855791 and rs4820268) and iron intake are associated with low iron and hemoglobin 19 concentration in under-two-year-old children.
The manuscript is understandable, well structured and all sections are interesting and relevant. However, the information provided to support the arguments being made should be increased because the manuscript features only 15 references and its length is too short.
It would be recommendable to include, if possible, additional parameters related to iron metabolism and increase the length of the introduction and discussion. The conclusions of the paper are clearly stated and adequately tie together the other elements of the manuscript.
Minor comments:
-Line 119: "and borderline significantly associated with lower SF by 4.50 μg/L of rs855791(p = 0.080) (Table 3)". There is no such association, because it there are no statistical signification. please rephrase.
-Table2: please replace "Energi" by "Energy".
-Line 170. Please replace "linier" by "linear"
-Line 201-202: The sentence "In summary our study showed that amongst these under two children where anemia and iron deficiency are prevalent...." is confusing and not understandable for the reader, please rewrite it.
Author Response
Reviewer 3
1. The manuscript is understandable, well-structured and all sections are interesting and relevant. However, the information provided to support the arguments being made should be increased because the manuscript features only 15 references and its length is too short. It would be recommendable to include, if possible, additional parameters related to iron metabolism and increase the length of the introduction and discussion. The conclusions of the paper are clearly stated and adequately tie together the other elements of the manuscript.
We already added body iron store parameter by calculation of serum ferritin and soluble sTfR following the formula body iron (mg/kg=-[log(R/F ratio)-2.8229]/0.1207 to lengthen the discussion.
· Djames, Cook, Hcarol, Flowers, SSBarry. The quantitative assessment of body iron. The American Society of Hematology (Blood Journal). 2003. (101): 9. 3359-63 p.
2. Line 119: "and borderline significantly associated with lower SF by 4.50 μg/L of rs855791(p =0.080) (Table 3)". There is no such association, because it there are no statistical signification. please rephrase.Re: This has been revised.
3. Table2: please replace "Energi" by "Energy".
Re: This has been revised.
4. Line 170. Please replace "linier" by "linear"
Re: This has been revised.
5. Line 201-202: The sentence "In summary our study showed that amongst these under two children where anemia and iron deficiency are prevalent...." is confusing and not understandable for the reader, please rewrite it.
Re: We have rewritten this into “In summary our study showed that amongst these under two children where anemia and iron deficiency are prevalent, variant allele G at rs4820268 of TMPRSS6 gene respectively was associated with lower SF concentration; but the association is still weaker than that of iron intake”.